# Biopharmaceutical Evaluation of Capsules with Lyophilized Apple Powder

**DOI:** 10.3390/molecules26041095

**Published:** 2021-02-19

**Authors:** Aurita Butkevičiūtė, Mindaugas Liaudanskas, Kristina Ramanauskienė, Valdimaras Janulis

**Affiliations:** 1Department of Pharmacognosy, Lithuanian University of Health Sciences, Sukileliu av. 13, LT-50162 Kaunas, Lithuania; Mindaugas.Liaudanskas@lsmu.lt (M.L.); Valdimaras.Janulis@lsmuni.lt (V.J.); 2Department of Clinical Pharmacy, Lithuanian University of Health Sciences, Sukileliu av. 13, LT-50162 Kaunas, Lithuania; Kristina.Ramanauskiene@lsmuni.lt

**Keywords:** apple, phenolic compounds, dissolution test, HPLC-DAD

## Abstract

Apples are an important source of biologically active compounds. Consequently, we decided to model hard gelatin capsules with lyophilized apple powder by using different excipients and to evaluate the release kinetics of phenolic compounds. The apple slices of “Ligol” cultivar were immediately frozen in a freezer (at −35°C) with air circulation and were lyophilized with a sublimator at the pressure of 0.01 mbar (condenser temperature, −85°C). Lyophilized apple powder was used as an active substance filled into hard gelatin capsules. We conducted capsule disintegration and dissolution tests to evaluate the quality of apple lyophilizate-containing capsules of different encapsulating content. Individual phenolic compounds can be arranged in the following descending order according to the amount released from the capsules of different compositions: chlorogenic acid > rutin > avicularin > hyperoside > phloridzin > quercitrin > (−)-epicatechin > isoquercitrin. Chlorogenic acid was the compound that was released in the highest amounts from capsules of different encapsulating content: its released amounts ranged from 68.4 to 640.3 μg/mL. According to the obtained data, when hypromellose content ranged from 29% to 41% of the capsule mass, the capsules disintegrated within less than 30 min, and such amounts of hypromellose did not prolong the release of phenolic compounds. Based on the results of the dissolution test, the capsules can be classified as fast-dissolving preparations, as more than 85% of the active substances were released within 30 min.

## 1. Introduction

Apples are among the most consumed fruits in Lithuania and worldwide. According to the data of 2020, 84.63 million tons of apples were grown in the world [1]. The greatest amounts of apples are grown in China (around 40.92 million tons). In the US, around 5.19 million tons are grown per year, and in Poland, the amount of apples grown per year is around 3.20 million tons [1]. Apples are widely used in the food industry in the production of various products and beverages (juices, wines, and ciders), and are also used unprocessed [2,3].

In the healthy food chain, apples are an important source of biologically active compounds [4]. They have been found to contain a complex of phenolic (8.2–360.75 mg/g^−1^) [5], and triterpenic (0.47–3.75 mg/g^−1^) [6] compounds, glucose (1.96–75.91 mg/g^−1^), fructose (15.52–342.14 mg/g^−1^), sucrose (11.81–43.79 mg/g^−1^), xylitol (1.01–13.79 mg/g^−1^), pectins (4.05–19.72 mg/g^−1^), organic acids (malic acid (38.96–107.28 mg/g^−1^), citric acid (1.84–4.12 mg/g^−1^), maleic acid (0.17–0.23 mg/g^−1^), pyruvic acid (0.14–0.34 mg/g^−1^), shikimic acid (0.04–0.17 mg/g^−1^)) [7,8], vitamins (21.08 mg/g^−1^) [9], macroelements (K (1.07–1.12 mg/g^−1^), P (0.074–0.11 mg/g^−1^), Mg (0.05–0.08 mg/g^−1^), Ca (0.04–0.06 mg/g^−1^), Na (0.007–0.04 mg/g^−1^)), microelements (Fe (0.001–0.003 mg/g^−1^), Zn (0.0004–0.019 mg/g^−1^), Mn (0.0003–0.0004 mg/g^−1^), Cu (0.0003–0.0005 mg/g^−1^) [10], and fiber (3700–4500 mg/g^−1^) [5].

Biologically active substances in apples affect the biological systems of the human body. They have an antioxidant effect, neutralizing harmful reactive oxygen and nitrogen species that cause structural damage to the body’s molecules, which, in turn, is directly linked to the development and progression of numerous diseases (cardiovascular and neurodegenerative diseases, cancer, diabetes, etc.) [11,12]. The biologically active compounds accumulated in apples are potentially valuable for the prevention of various diseases [13,14]. During the recent period, in order to preserve the chemical composition of high-quality fruit, lyophilization is increasingly used in the food industry. Lyophilized apples with a chemical composition identical to that of fresh apples can be used in the functionalization of food supplements and various pharmaceutical dosage forms.

One important step in modeling a solid pharmaceutical dosage form is the selection of the category of the pharmaceutical form. Hard capsules are a convenient and widely used dosage form in which drugs and excipients content are placed in the capsules. One of the most widely used capsules may not be resistant to and dissolved in gastric juice, so we decided to use this capsules type in the study. In addition, the capsules may be resistant to gastric juice and dissolved in a selected part of the gastrointestinal tract if appropriate excipients are selected. Therefore, one of the most important advantages of capsules is that their use allows for the localization of the gastrointestinal tract site of the release of biologically active compounds from the capsule, thus helping to protect the acid-sensitive biologically active substances from the destructive effects of gastric juice [15].

Another equally important property of hard capsules is that the dissolution kinetics of the active substances can be modified by using different excipients in the modeling of hard capsules. In our study, the active substance in capsules is a lyophilized apple powder rich in phenolic compounds. Excipients are used in the manufacturing process of encapsulated mixtures in order to increase the bulk of the encapsulated substances, to reduce their adhesion, to improve flowability, and/or to promote disintegration or penetrability to water, thus differentially modifying the release of the active ingredients from the capsule. The use of excipients can slow down the release of the capsule contents, thus ensuring a longer and more sustained effect of the biologically active substances in the capsule [15,16]. It is important to select suitable excipients that would ensure adequate dissolution kinetics of the medicinal substance. Microcrystalline cellulose, starch, and D (+)-glucose were used as fillers in the encapsulating mixture. Hypromellose was selected as a drug release modifier that would prolong the release of the active ingredient. Silicon dioxide was selected as the tackifier of the encapsulated mixture.

About 50% of the currently biopharmaceuticals are lyophilized, representing the most common formulation strategy [17]. Lyophilization or freeze drying is a process in which water is frozen, followed by its removal from the sample, initially by sublimation (primary drying) and then by desorption (secondary drying). Freeze-drying is a process of drying in which water is sublimed from the product after it is frozen [17,18]. Although on the pharmacy there are quite a number of medicines, and in particular food supplements, which use various plant extracts as the active ingredients in hard capsules, the use of lyophilized fruits and botanical raw materials in the production of hard capsules is still a rare phenomenon. Lyophilized fruits (apples, pears, plums, peaches), berries (blueberries, cranberries, strawberries), and vegetables (carrots, beets) are used in the food industry and can be added to various types of cocktails, cereals, drinks, meals, juices, yogurts, ice cream; freeze-dried fruit powder can also be added to food supplements thus increasing the content of biologically active compounds [19,20,21,22]. Dehydrated, freeze-dried fruit, berries, and vegetables promote long-term storage of freeze-dried food or food supplements. Lyophilization preserves most of the biologically active compounds stored in the fruit, while other methods often destroy thermolabile compounds. Lyophilization does not involve the use of chemical substances, which is crucial for providing the consumers with safe, effective, and environmentally friendly products or food supplements with the highest content of biologically active compounds.

The aim of our study was to model hard gelatin capsules with lyophilized apple powder by using different excipients and to evaluate the release kinetics of phenolic compounds.

## 2. Results and Discussion

### 2.1. Qualitative and Quantitative Analysis of Phenolic Compounds of Apple Lyophilisate

During the first stage of the study, the quantitative composition of the lyophilized apple powder was analyzed. Ethanol extracts of the lyophilized apple powder were analyzed by applying high performance liquid chromatography (HPLC). The obtained results allowed for an accurate assessment of the qualitative and quantitative composition of individual phenolic compounds in the studied apple lyophilisate powder. Different groups of phenolic compounds were identified and quantified in the analyzed apple lyophilisate: quercetin glycosides (rutin, hyperoside, isoquercitrin, reynoutrin, avicularin, and quercitrin), flavan-3-ols (procyanidin B1, procyanidin B2, procyanidin C1, (+)-catechin, and (−)-epicatechin), dihydrochalcones (phloridzin), and phenolic acids (chlorogenic acid). The data obtained by Jakobek et al. corroborate to the results of our research [23]. The chromatogram of the tested apple lyophilisate ethanol extract is shown in Figure 1.

The sum of the identified and quantified individual phenolic compounds found in the apple lyophilisate of the “Ligol” cultivar was 1.94 ± 0.05 mg/g. Studies of the fruit samples of apple cultivars grown in the orchards of the Marche region of Italy showed that the total amount of phenolic compounds ranged from 0.82 to 3.60 mg/g and confirm the results of our research [5]. Chlorogenic acid predominated among all the identified phenolic compounds. Its content was 0.67 ± 0.10 mg/g, which accounted for 34.5% of the total content of all the detected phenolic compounds (Figure 2).

The results of this study confirmed those of the previous studies reporting that chlorogenic acid is one of the most predominant components in apples [24,25]. The amount of chlorogenic acid in apple samples grown in Italy was found to vary from 0.12 to 0.63 mg/g [26]. In our study, a higher amount of chlorogenic acid compared with the amount described by Italian scientists was determined. Chlorogenic acid has properties important for human health, such as antioxidant activity [27,28], anti-inflammatory activity [29,30], the reduction in the risk of type 2 diabetes [31], the improvement of cardiovascular function [32,33], and the inhibition of the processes of carcinogenesis [34,35,36].

Another group of flavan-3-ol compounds with diverse biological activity identified in the apple lyophilisate consisted of monomeric compounds ((+)-catechin and (−)-epicatechin) and oligomeric compounds (procyanidin B1, procyanidin B2, and procyanidin C1). The total amount of compounds in flavan-3-ol group was 0.82 ± 0.03 mg/g, which accounted for 42.3% of the total amount of the identified phenolic compounds (Figure 2). The flavan-3-ol content in fruit samples of apple cultivars grown in Croatian orchards was found to vary from 0.02 to 0.69 mg/g [23]. We established the higher amount of total flavan-3-ols compared with the amount determined by Croatian scientists. The variability in the quantitative composition of the individual compounds of the flavan-3-ol group in the apple lyophilisate is shown in Figure 2. The predominant compounds of the flavan-3-ol group in the apple lyophilisate was procyanidins. The highest amount of procyanidin B2 and procyanidin C1 were determined as 0.30 ± 0.08 mg/g and 0.23 ± 0.06 mg/g, respectively (Figure 2). The amount of procyanidin B2 and procyanidin C1 in fruit samples of apple cultivars grown in Polish orchards ranged from 0.07 to 2.00 mg/g and 0.0006 to 0.97 mg/g, accordingly [37]. Our study results confirmed Polish research results. Procyanidins are important for the human body as they exhibit antioxidant, anticancer, anti-inflammatory, platelet aggregation-reducing, and cholesterol-reducing effects [38,39,40]. Monomeric flavan-3-ols, which together with procyanidins may be responsible for the cholesterol-lowering [41] and vasodilating [42] effect of the apples, also inhibit sulfotransferases, and thus can regulate the biological activity of hydroxysteroids and can act as natural chemopreventive agents [43].

Phloridzin, a compound of the dihydrochalcone group, was detected in the apple lyophilisate as well. Its quantitative content in the sample was 0.08 ± 0.005 mg/g, which accounted for 4.1% of the total amount of phenolic compounds detected in the apple lyophilisate (Figure 2). The amount of phloridzin in fruit samples of apple cultivars grown in orchards in the Garfagnana region of Italy ranged from 0.01 to 0.05 mg/g [44]. The results obtained in these studies confirm the results obtained in our research. Phloridzin exhibits important antidiabetic activity [45,46], and therefore apples, as botanical raw materials accumulating this compound, can be potentially useful for the prevention of diabetes mellitus [14].

The qualitative and quantitative composition of quercetin glycosides is shown in Figure 3. The total amount of quercetin glycosides was 0.37 ± 0.12 mg/g, which accounted for 19.1% of the total amount of phenolic compounds detected in the apple lyophilisate.

Studies of fruit samples of apple cultivars grown in Croatian orchards showed that quercetin glycosides levels ranged from 0.20 to 1.22 mg/g [23]. The results of our study are corroborated by research data obtained by Croatian researchers. Hyperoside predominated among all the identified quercetin glycosides. Its content was 0.12 ± 0.06 mg/g, which accounted for 6.2% of the total content of all the detected phenolic compounds (Figure 3). The amount of hyperoside in fruit samples of apple cultivars grown in Italian orchards ranged from 0.0003 to 0.002 mg/g [47]. The apple lyophilisate contained higher amounts of hyperoside compared to these found in fruit samples of apple cultivars grown in Italian orchards. The amount of quercitrin was 0.09 ± 0.04 mg/g and these results confirmed Italian study results that report that the amount of quercitrin found in apple samples ranged from 0.005 to 0.043 mg/g [47].

All the quercetin glycosides identified and quantified in the ethanol extract of the apple lyophilisate can be ranked in the following ascending order by their content: rutin < isoquercitrin < reynoutrin < avicularin < quercitrin < hyperoside. Hyperoside was a predominant component among quercetin glycosides in the ethanol extracts of the fruit samples of apple cultivars selected for this study. Rutin was the minor component among all the quercetin derivatives. These results are consistent with those of the previously published studies, which reported hyperoside to be one of the predominant compounds [3,48].

### 2.2. Biopharmaceutical Evaluation of Hard Gelatin Capsules

Following the analysis of the composition of biologically active compounds in the apple lyophilisate, the selected amount of the active substance was 0.100 g of lyophilized apple powder per capsule. During the next stage of the research, the selection of excipients was performed. Microcrystalline cellulose, starch, and glucose were chosen as fillers to increase the mass of the encapsulated content when modeling apple lyophilisate-containing capsules (Table 1).

Microcrystalline cellulose, one of the most commonly used excipients in the manufacture of solid dosage forms, has excellent compressibility and disintegration-enhancing properties [49]. Glucose is well soluble in water and has good sensory properties, but is hygroscopic [50]. Hypromellose was included in the composition of the capsules to prolong their disintegration time and to extend their therapeutic efficacy [51]. All the compositions of the apple lyophilisate-containing capsules are presented in Table 1.

The quality of the manufactured capsules was evaluated according to the following parameters: uniformity of the capsule mass, capsule disintegration time, and the amount of the active substances released from the capsule (the dissolution test). Data presented in Table 1 shows that the indices of the uniformity of the capsule mass were similar. Thus, it can be stated that the selected excipients and their amounts ensured accurate dosage of the encapsulated mixture into the capsules. During the next stage of the study, capsule disintegration and dissolution tests were performed. Based on the results of these tests, the bioavailability of the modeled capsules can be predicted. The results of these tests are presented in Figure 4 and Table 2.

Disintegration time is an important indicator in assessing the quality of capsules. The release of the drug is known to begin only after the disintegration of the capsule. The excipients in the encapsulated mixture must not interfere with the solubility and disintegration of the capsule. According to the test results presented in Figure 4, the capsules of the compositions N1–N9 disintegrated within less than 15 min, the capsules of the composition N10 disintegrated within 20 min, while the capsules of the composition N11 did not disintegrate after more than 60 min (Figure 4). The European Pharmacopoeia states that non-modified capsules must disintegrate within 30 min, and thus it can be stated that capsules N1–N10 meet the requirements of the European Pharmacopoeia [52]. The results of our study were confirmed by the results described by other scientists. Vyas et al. showed that lyophilized *Vasa Swaras* capsules undergo disintegration in more than 4.23 min [53]. Esmaeili et al. confirmed our study results, that hard gelatin capsules from *Pinus eldarica* bark extract disintegrate between 9.4 and 20.0 min [54]. There was no statistically significant difference (*p* > 0.05) between the disintegration time of the N1–N9 capsules compared to the disintegration time of the other tested capsules. The results of the testing showed that capsules disintegrated more slowly when hypromellose was used as a filler (Figure 4). A statistically significant difference (*p* < 0.05) was found between the disintegration time of the N10 capsules and the disintegration time of the other tested capsules. In capsules of this composition, hypromellose made up about 71.4% of the capsule mass. This confirms the literature data indicating that hypromellose has disintegration-prolonging properties [51].

Dissolution test is one of the most important tests used for capsule quality assessment. The dissolution test evaluates the time taken for a defined amount or part of a drug to be released from a dosage form into a solution. During the test, it is important to choose the right dissolution medium. For this reason, the next stage of the study focused on the selection of a suitable solvent for the active substances. Table 2 presents test results showing that the maximum amount of the active ingredients was released from the capsule when a 1:1 mixture of ethanol and water was used as the solvent.

According to the amount of the active substance released from the capsules of the compositions N1–N11, individual phenolic compounds can be arranged in the following descending order: chlorogenic acid > rutin > avicularin > hyperoside > phloridzin > quercitrin > (−)-epicatechin > isoquercitrin (Table 2). Based on the results of the study, chlorogenic acid was the compound that was released in the highest amounts from capsules N1–N11, its released amount ranged from 68.4 to 640.3 μg/mL (Table 2).

The results of the testing showed that the selected excipients did not affect the active ingredients because all the capsule formulations released the compounds that were detected in the apple lyophilisate. The amounts of excipients affected the dissolution kinetics of the modeled capsules. The results showed in Table 3 reveals that unequal amounts of the phenolic compounds were released after 30 min.

There was no statistically significant difference (*p* > 0.05) between the amount of phenolic compounds released from the N1–N7 capsules after 30 min (Table 3). Thus, it can be argued that the amount of hypromellose used in capsules N1–N7 did not prolong the release of the active substance [55]. The results of the testing revealed that the amount of hypromellose used in capsule production did not prolong the release of the active substance when its amount in the capsules ranged from 29% to 41%. The selected excipients with different properties such as glucose, microcrystalline cellulose and starch did not affect the release of the active ingredients, and no statistically significant (*p* > 0.05) difference was found between the amount of the active compounds released from capsules N3, N4, and N5.

The results of the investigation exposed that the higher amount of hypromellose ranging from 50% to 83% in capsules did prolong the release of the active substance. The results show that after 30–60 min the lowest amount of the phenolic compounds is released from capsule N11 containing 0.500 g of the hypromellose. Of studied N8–N11 composition capsules was released than 85% of active compounds after 75 min (Table 4).

There was no statistically significant difference determined between the amount of released active compounds of capsules N8–N11 after 75 and 90 min (*p* > 0.05). The testing results confirmed the literature data indicating that the dissolution test is an informative tool for assessing the quality of solid dosage forms, helping to evaluate how the dosage form releases the active ingredients [56]. The results of the study showed that the selected excipients were suitable for the modeling of capsules with lyophilized apple powder. The results of this study proved that hypromellose is an appropriate excipient for prolonging the release of an active substance. Active compounds release and disintegration rate depends on the quantity of hypromellose in a capsule. The corresponding amount of hypromellose in capsule N1–N7 neither prolonged the disintegration of the capsules nor affected the dissolution rate of the active compounds. The results of the study showed that the use of hypromellose as a filler in higher amounts (N8–N11) prolongs the release of the active compounds.

The literature has demonstrated concerning data related to herbal medicinal product dissolution tests. A study performed with phytomedicines based on *Ginkgo biloba* extract capsules and tablets found marked differences in dissolution behavior were established, with values of 99% and 33% dissolution, on average, after 15 min and 60 min, respectively [57]. The dissolution profile of *Senna* sp. was less than 10% sennoside release from capsules containing dry extract, in a period of 60 min, in contrast to lyophilized *Senna* sp. extract, which attained around 90% dissolution after the same period. Data on the dissolution test of *Passiflora* sp. showed that capsules containing the crude extract presented 50% dissolution while lyophilized extract and other standardized extract reached around 100% for the same period [58].

The modeled capsules are suitable for internal use as a food supplement that contains all the components of the chemical composition of the phenolic compounds found in lyophilized apples, as these components have a wide range of biological effects on the human body. Given the fact that the bioavailability of phenolic compounds is not particularly high, compounds with a lower molecular weight are more readily absorbed in the gastrointestinal tract [59]. According to scientific literature, various groups of phenolic compounds are absorbed at a rate of 0.3–43%, and the metabolite content circulating in the plasma can be low [60]. Chlorogenic acid absorption is approximately 33%, and the majority of chlorogenic acid will reach the large intestine, while (+)-catechin and (−)-epicatechin are both absorbed by small intestinal epithelial cells [11]. The route of administration, the release of a dosage form, and absorption are known to affect bioavailability [61]. The bioavailability of flavonoids depends on their physicochemical properties. Flavonoids with complex structures and larger molecular weights, bioavailability may be even lower [62,63]. The absorption of flavonoids in the small intestine is limited owing to their molecular weight and hydrophilicity of their glycosides. It is therefore important to select suitable excipients that do not limit the therapeutic efficacy of the active compounds [64].

Since the active compounds in the apple lyophilisate were not highly soluble, we tried to model capsules whose excipients would not impair their solubility. When modeling capsules that contain active substances of botanical origin, it is expedient to avoid possible interactions between the lyophilized apple powder and the excipients, since dissolution, base structure, molecular size, and interaction with other components are the major physiochemical properties that lower the effectiveness of phenolic compounds [59]. According to the scientific literature, lyophilisate of *Vasa Swaras* capsules improved the stability and oral bioavailability of vasicine as well as reduced its conversion to vasicinone. The pharmacokinetic profiles of the various formulations inferred that the highest bioavailability of vasicine was obtained from hard gelatin capsules of lyophilized *Vasa Swaras* [53].

Considering the increasing consumption of herbal medicines, food supplements, there is a need for studies to ensure quality from raw material to finished product. Quality control is essential for the safety of functional food or nutrition supplements. Thus, biopharmaceutical studies of capsules with apple lyophilisate are of great importance since they can contribute toward ensuring future research for the planning and manufacture of innovative nutrition supplements.

## 3. Materials and Methods

### 3.1. Plant Materials

The study included “Ligol” apple cultivar (a winter cultivar bred in Poland). The apple trees were grown in the experimental orchard (block 2, row 4, trees 21–40) of the Institute of Horticulture, Lithuanian Research Centre for Agriculture and Forestry, Babtai, Lithuania (55°60′ N, 23°48′ E). The altitude of Babtai town is 57 m above sea level. Trees were trained as a slender spindle. Pest and disease management was carried out according to the rules of integrated plant protection. The experimental orchard was not irrigated. Tree fertilization was performed based on the results of soil and leaf analysis. Nitrogen was applied before flowering at the rate of 80 kg ha^−1^, and potassium was applied after the harvest at the rate of 90 kg ha^−1^. Soil conditions of the experimental orchard were the following: clay loam, pH—7.3, humus—2.8%, P_2_O_5_—255 mg kg^−1^, and K_2_O—230 mg kg^−1^. The apples harvested in September 2019 were immediately lyophilized and used for the study.

### 3.2. Chemicals and Solvents

All solvents, reagents, and standards used were of analytical grade. Acetonitrile, acetic acid, D(+)-glucose monohydrate, hypromellose, Aerosil^TM^ 200, microcrystalline cellulose, silicon dioxide and starch were obtained from Sigma-Aldrich GmbH (Buchs, Switzerland), and ethanol was obtained from Stumbras AB (Kaunas, Lithuania). Hyperoside, rutin, quercitrin, phloridzin, procyanidin B1, procyanidin B2, and chlorogenic acid standards were purchased from Extrasynthese (Genay, France), reynoutrin, (+)-catechin and (−)-epicatechin were purchased from Sigma-Aldrich GmbH (Buchs, Switzerland), and avicularin, procyanidin C1, and isoquercitrin were purchased from Chromadex (Santa Ana, USA). In this study, we used deionized water produced by the Milli-Q^®^ (Millipore, Bedford, MA, USA) water purification system.

### 3.3. Preparation of Apple Lyophilisate

The apples were cut into slices of equal size (up to 1 cm in thickness), and the stalks and the seeds were removed. The apple slices were immediately frozen in a freezer (at −35 °C) with air circulation. Apple samples were lyophilized with a ZIRBUS sublimator 3 × 4 × 5/20 (ZIRBUS technology, Bad Grund, Germany) at the pressure of 0.01 mbar (condenser temperature, −85 °C). The lyophilized apple slices were ground to fine powder (particle size about 100 µm) by using a knife mill Grindomix GM 200 (Retsch, Haan, Germany).

### 3.4. Preparation of Phenolic Extracts

During the analysis of phenolic compounds, 2.5 g of lyophilizate powder (exact weight) was weighed, added to 30 mL of 70% (*v*/*v*) ethanol, and extracted in a Sonorex Digital 10 P ultrasonic bath (Bandelin Electronic GmbH & Co. KG, Berlin, Germany) at room temperature for 20 min. The obtained extract was filtered through a paper filter, and the residue on the filter was washed with 70% (*v*/*v*) ethanol in a 50 mL flask until the extract volume was reached. The conditions of the extraction were chosen based on the results of the tests for setting the extraction conditions [65].

### 3.5. Qualitative and Quantitative Analysis by HPLC-PDA Method

The qualitative and quantitative HPLC analysis of phenolic compounds was performed with a Waters 2998 PDA detector (Waters, Milford, USA). Chromatographic separations were carried out by using a YMC-Pack ODS-A (5 μm, C18, 250 × 4.6 mm i.d.) column. The column was operated at a constant temperature of 25 °C. The volume of the analyzed extract was 10 μL. The flow rate was 1 mL/min. The mobile phase consisted of 2% (*v*/*v*) acetic acid (solvent A) and acetonitrile (solvent B). Gradient variation: 0–30 min 3–15% B, 30–45 min 15–25% B, 45–50 min 25–50% B, and 50–55 min 50–95% B. For the quantitative analysis, the calibration curves were obtained by injecting the known concentrations of different standard compounds. All the identified phenolic compounds were quantified at λ = 200–400 nm wavelength [66,67].

### 3.6. Encapsulation Process

Compositions of capsules fillings are given in Table 1. Powder was prepared by simple mixing of the mixture of apple lyophilizate with excipient. Filled capsules were prepared using the manual capsule filling machine (Capsuline, Davie, FL, USA).

#### 3.6.1. Test of the Uniformity of Mass of Single-Dose Preparations

The tested capsules were weighed, and the mean weight of 1 capsule was determined [68]. One capsule was weighed, apple lyophilizate and excipients mixture were poured out, and then the capsule shell was weighed. Subsequently, the mass of the content—i.e., the difference between the weight of the capsule and the weight of the shell—was calculated. This procedure was applied to each modeled capsule. The allowed deviation for capsules weighing not more than 250 mg was 7.5%.

#### 3.6.2. Capsule Disintegration Test

The capsule disintegration time was determined based on the methodology outlined in Ph. Eur. 2.9.1. [69]. The device C-MAG HS7 (IKA^®^-Werke GmbH & Co, Staufen, Germany) was used to determine the disintegration time. The disintegration medium was 0.1 M hydrochloric acid solution, temperature 37 ± 0.50 °C, observed for 30 min.

#### 3.6.3. Capsule Dissolution Test

The capsule dissolution test was performed using a Sotax AT 7smart dissolution tester (SOTAX AG, Allschwil, Switzerland). The acceptor medium was an ethanol-water mixture at the ratio of 1:1, the temperature being 37.0 ± 0.5 °C. The volume of the medium was 250 mL. The samples were taken after 15, 30, 60, 75 and 90 min. The sample volume was 10 mL. The analysis of the active compounds was performed by applying HPLC.

### 3.7. Statistical Analysis

The statistical analysis of the study data was performed by using Microsoft Office Excel 2013 (Microsoft, Redmond, WA, USA) and SPSS 25.0 (SPSS Inc., Chicago, IL, USA) computer software. All the results obtained during the HPLC analysis were presented as means of three consecutive test results and standard deviations. To evaluate the variance in the quantitative composition, we calculated the coefficient of variation. Univariate analysis of variance (ANOVA) was applied in order to determine whether the differences between the compared data were statistically significant. The hypothesis about the equality of variances was verified by applying Levine’s test. If the variances of independent variables were found to be equal, Tukey’s multiple comparison test was used. The differences were regarded as statistically significant at *p* < 0.05.

## 4. Conclusions

Excipients and their amounts may affect the dissolution kinetics of the phenolic compounds. Hypromellose prolonged the disintegration time of the modeled capsules when its amount reached 50–83% of the capsule weight. The selected fillers did not affect the kinetics of the release of the phenolic compounds from the capsules. Based on the results of the dissolution test, the capsules can be classified as fast-dissolving preparations since more than 85% of the active substance was released within 30 min.

According to the amount released from the capsules of different encapsulating content, individual phenolic compounds can be arranged in the following descending order: chlorogenic acid > rutin > avicularin > hyperoside > phloridzin > quercitrin > (−)-epicatechin > isoquercitrin. Chlorogenic acid was the compound that was released in the highest amounts from capsules of different encapsulating content: its released amounts ranged from 68.4 to 640.3 μg/mL.

The results of the solubility and disintegration tests proved that the capsules of the proposed composition are appropriate for internal use. The proposed product could serve as a basis for the development of food supplements with lyophilized apple powder.

## Figures and Tables

**Figure 1 molecules-26-01095-f001:**
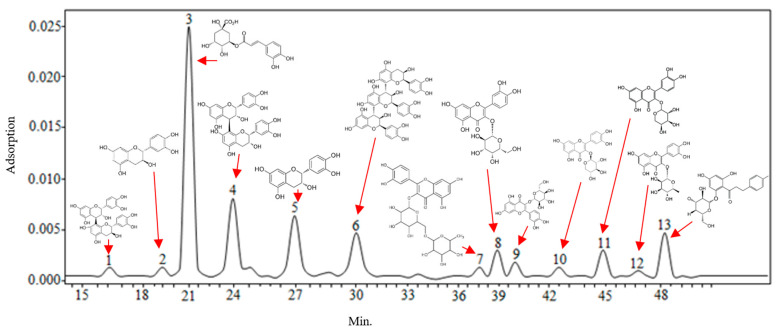
Chromatogram of the ethanol extract of the apple lyophilisate. Analytes determined at λ = 280 nm wavelength: 1—procyanidin B1; 2—(+)-catechin; 3—chlorogenic acid; 4—procyanidin B2; 5—(−)-epicatechin; 6—procyanidin C1; at λ = 360 nm wavelength: 7—rutin; 8—hyperoside; 9—isoquercitrin; 10—reynoutrin; 11—avicularin; 12—quercitrin; at λ = 280 nm wavelength: 13—phloridzin.

**Figure 2 molecules-26-01095-f002:**
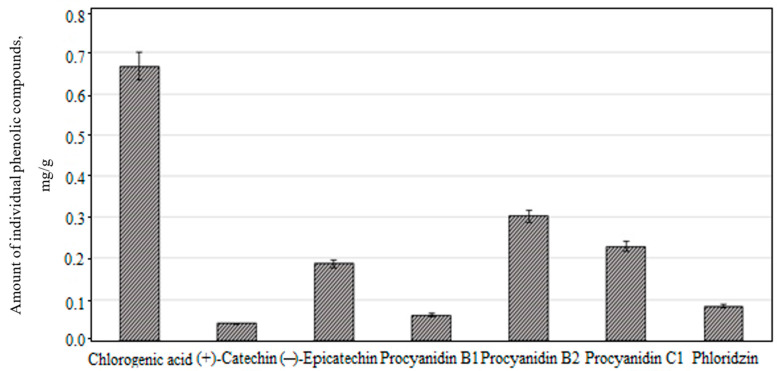
The amount of individual flavan-3-ols, phloridzin, and chlorogenic acid in ethanol extracts obtained from the apple fruit of the “Ligol” cultivar grown in Lithuania.

**Figure 3 molecules-26-01095-f003:**
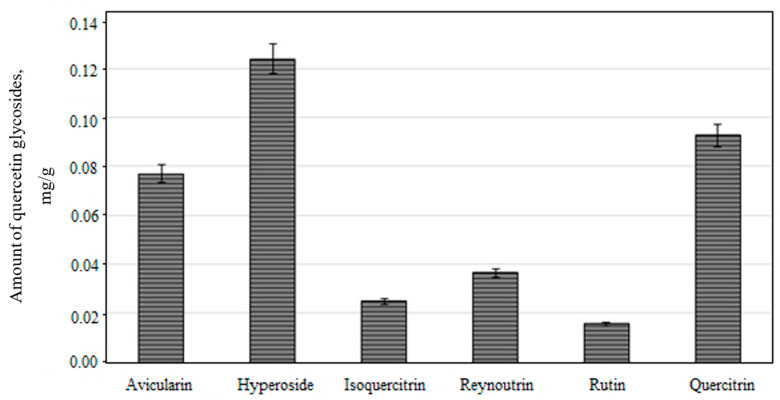
Concentration of individual quercetin glycosides in ethanol extracts obtained from the fruit of the “Ligol” apple cultivar grown in Lithuania.

**Figure 4 molecules-26-01095-f004:**
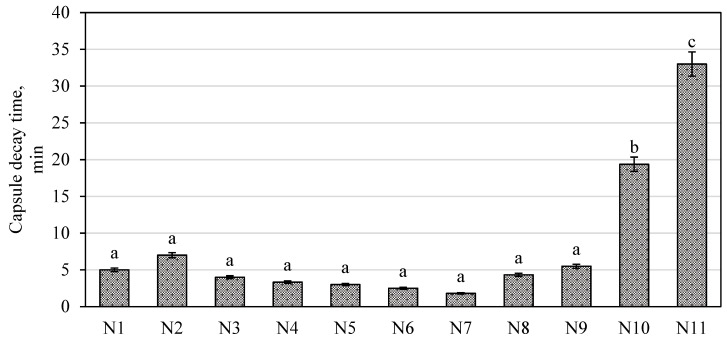
Results of the capsule disintegration test. The means followed by different letters are significantly different at *p* < 0.05.

**Table 1 molecules-26-01095-t001:** Compositions of the apple lyophilisate-containing capsules.

CC	AL, g	SDX, g	MCC, g	HPMC, g	HEC, g	GL, g	ST, g	TCM, g	FQ	MM, g
N1	0.100	0.001	0.019	0.050	-	-	-	0.170	The mass is powdery, the capsule is filled completely	0.171
N2	0.100	0.001	-	0.069	-	-	-	0.170		0.171
N3	0.100	0.001	-	-	-	0.069	-	0.170		0.069
N4	0.100	0.001	0.069	-	-	-	-	0.170		0.171
N5	0.100	0.001	-	-	-	-	0.069	0.170		0.171
N6	0.100	-	-	0.070	-	-	-	0.170		0.172
N7	0.100	-	-	-	0.070	-	-	0.170		0.171
N8	0.100	-	-	0.100	-	-	-	0.200		0.202
N9	0.100	-	-	0.150	-	-	-	0.250		0.251
N10	0.100	-	-	0.250	-	-	-	0.350		0.349
N11	0.100	-	-	0.500	-	-	-	0.600		0.658

CC = capsule compositions; AL = apple lyophilisate; SDX = silicon dioxide; MCC = microcrystalline cellulose; HPMC = hydroxypropyl methylcellulose; HEC = hydroxyethyl cellulose; GL = glucose; ST = starch; TCM = theoretical capsule mass; FQ = filling quality; MM = mean mass.

**Table 2 molecules-26-01095-t002:** Results of the in vitro dissolution test after 30 min. The means followed by different letters in the columns are significantly different at *p* < 0.05.

	Chlorogenic Acid	Rutin	Hyperoside	Isoquercitrin	Quercitrin	Avicularin	(−)-Epicatechin	Phloridzin
CC	μg/mL							
N1	372.5 ^B^	160.0 ^B^	144.5 ^B^	7.0 ^A^	97.5 ^B^	149.0 ^B^	17.5 ^A^	122.5 ^B^
N2	367.5 ^B^	157.5 ^B^	140.0 ^B^	6.0 ^A^	89.5 ^B^	142.5 ^B^	9.5 ^B^	117.5 ^B^
N3	372.5 ^B^	149.5 ^B^	139.0 ^B^	7.5 ^A^	91.5 ^B^	148.5 ^B^	10.0 ^B^	121.5 ^B^
N4	367.5 ^B^	152.5 ^B^	140.5 ^B^	5.5 ^A^	92.5 ^B^	142.0 ^B^	14.5 ^A^	125.0 ^B^
N5	385.0 ^B^	155.0 ^B^	132.5 ^B^	4.9 ^A^	87.5 ^B^	141.5 ^B^	8.5 ^B^	110.0 ^B^
N6	589.5 ^A^	265.6 ^A^	240.4 ^A^	5.0 ^A^	170.7 ^A^	253.7 ^A^	6.3 ^B,C^	140.5 ^A^
N7	640.3 ^A^	268.3 ^A^	241.8 ^A^	4.9 ^A^	166.5 ^A^	254.3 ^A^	10.2 ^B^	119.9 ^B^
N8	167.9 ^C^	62.1 ^C^	60.9 ^C^	2.3 ^B^	40.3 ^C^	61.8 ^C^	1.2 ^D^	50.6 ^C^
N9	128.3 ^C^	60.8 ^C^	59.6 ^C^	2.3 ^B^	39.4 ^C^	59.6 ^C^	1.1 ^D^	49.5 ^C^
N10	126.4 ^C^	39.9 ^C^	49.4 ^C^	2.9 ^B^	33.3 ^C^	50.4 ^C^	3.8 ^C^	40.9 ^C^
N11	68.4 ^D^	32.4 ^D^	31.7 ^D^	1.2 ^C^	21.0 ^D^	31.8 ^D^	0.6 ^D^	26.4 ^D^

CC = capsule compositions.

**Table 3 molecules-26-01095-t003:** Results of the dissolution test for phenolic compounds released from the N1–N11 capsules after 30 min. The means followed by different letters in the columns are significantly different at *p* < 0.05.

Release Content, %	Chlorogenic Acid	Rutin	Hyperoside	Isoquercitrin	Quercitrin	Avicularin	(−)-Epicatechin	Phloridzin
N1	96.0 ^A^	94.0 ^A^	88.0 ^A^	85.0 ^A^	80.0 ^A^	90.0 ^A^	83.0 ^A^	97.0 ^A^
N2	94.0 ^A^	92.0 ^A^	87.0 ^A^	84.0 ^A^	82.0 ^A^	92.0 ^A^	80.0 ^A^	95.0 ^A^
N3	95.0 ^A^	85.0 ^A^	89.0 ^A^	81.0 ^A^	80.0 ^A^	90.0 ^A^	81.0 ^A^	94.0 ^A^
N4	97.0 ^A^	90.0 ^A^	89.0 ^A^	83.0 ^A^	85.0 ^A^	88.0 ^A^	83.0 ^A^	92.0 ^A^
N5	94.0 ^A^	88.0 ^A^	90.0 ^A^	85.0 ^A^	80.0 ^A^	92.0 ^A^	85.0 ^A^	96.0 ^A^
N6	95.0 ^A^	96.0 ^A^	90.0 ^A^	84.0 ^A^	84.5 ^A^	92.0 ^A^	81.0 ^A^	94.0 ^A^
N7	90.0 ^A^	92.0 ^A^	88.0 ^A^	80.0 ^A^	83.0 ^A^	91.0 ^A^	82.0 ^A^	92.0 ^A^
N8	55.0 ^B^	52.0 ^B^	53.0 ^B^	35.0 ^B^	38.0 ^B^	47.0 ^B^	35.0 ^B^	50.0 ^B^
N9	52.0 ^B^	55.0 ^B^	56.0 ^B^	34.0 ^B^	37.0 ^B^	40.0 ^B^	34.0 ^B^	54.0 ^B^
N10	50.0 ^B^	39.0 ^B^	42.0 ^B^	29.0 ^B^	30.0 ^B^	40.0 ^B^	29.0 ^B^	47.0 ^B^
N11	29.0 ^C^	26.0 ^C^	29.0 ^C^	19.0 ^C^	18.0 ^C^	27.0 ^C^	19.0 ^C^	28.0 ^C^

**Table 4 molecules-26-01095-t004:** Results of the dissolution test. Released content of the individual phenolic compounds after 60, 75, 90 min. The means followed by different letters in the columns are significantly different at *p* < 0.05.

Release Content, %	Chlorogenic Acid	Rutin	Hyperoside	Isoquercitrin	Quercitrin	Avicularin	(−)-Epicatechin	Phloridzin
After 60 min								
N8	88.0 ^A^	85.0 ^A^	83.0 ^A^	76.0 ^A^	80.0 ^A^	83.0 ^A^	81.0 ^A^	86.0 ^A^
N9	82.0 ^A^	82.0 ^A^	86.0 ^A^	79.0 ^A^	80.0 ^A^	81.0 ^A^	79.0 ^A^	82.0 ^A^
N10	79.0 ^A^	74.0 ^A^	76.0 ^A^	70.0 ^A^	73.0 ^A^	77.0 ^A^	69.0 ^A^	78.0 ^A^
N11	57.0 ^B^	55.0 ^B^	51.0 ^B^	45.0 ^B^	40.0 ^B^	57.0 ^B^	39.0 ^B^	55.0 ^B^
After 75 min								
N8	90.0 ^A^	86.0 ^A^	87.0 ^A^	81.0 ^A^	82.0 ^A^	87.0 ^A^	83.0 ^A^	88.0 ^A^
N9	91.0 ^A^	89.0 ^A^	88.0 ^A^	81.0 ^A^	83.0 ^A^	88.0 ^A^	82.0 ^A^	90.0 ^A^
N10	88.0 ^A^	88.0 ^A^	86.0 ^A^	83.0 ^A^	82.0 ^A^	87.0 ^A^	81.0 ^A^	89.0 ^A^
N11	92.0 ^A^	86.0 ^A^	87.0 ^A^	85.0 ^A^	83.0 ^A^	87.0 ^A^	81.0 ^A^	88.0 ^A^
After 90 min								
N8	95.0 ^A^	92.0 ^A^	93.0 ^A^	89.0 ^A^	88.0 ^A^	91.0 ^A^	88.0 ^A^	96.0 ^A^
N9	97.0 ^A^	94.0 ^A^	96.0 ^A^	92.0 ^A^	90.0 ^A^	93.0 ^A^	91.0 ^A^	96.0 ^A^
N10	96.0 ^A^	95.0 ^A^	90.0 ^A^	89.0 ^A^	87.0 ^A^	90.0 ^A^	89.0 ^A^	97.0 ^A^
N11	94.0 ^A^	93.0 ^A^	92.0 ^A^	88.0 ^A^	89.0 ^A^	91.0 ^A^	88.0 ^A^	93.0 ^A^

## Data Availability

All datasets generated for this study are included in the article.

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
