# Peer review of "Biopharmaceutical Evaluation of Capsules with Lyophilized Apple Powder"

_molecules, 2021, doi:10.3390/molecules26041095_

Round 1
Reviewer 1 Report
Dear Authors of manuscript entitled “Modeling and biopharmaceutical evaluation of capsules with lyophilized apple powder”,
below please find my comments and suggestions:
Abstract - please indicate strongly the aim of your research. In addition, please add information on how the lyophylisate was obtained and what the capsule was made of (it has a big impact on the results, so it is worth mentioning it at the beginning). Writing "capsules of different compositions" is too generate even for Abstract.
Maybe it would be helpful for potential readers to add kay words such as "lyophylysat" and/or "capsule"?
Introduction - More recent reports (than from 2014, L31) on the cultivation and consumption of apples are necessary
L 38-39, please provide the average amount/concentration of mentioned compounds.
Apples are great and we should eat them, but maybe it is worth mentioning that eating these fruits in excess is not advisable and is associated with unpleasant ailments? (fermentation, flatulence). Maybe this suggests that it would also be good to consume a lyophilisate, e.g. for someone who has flatulence after eating apples? Please treat this comment as a soft, non-binding remark.
L51-52, in my opinion the sentence: “In our study, we chose capsules as a solid pharmaceutical form” is useless in this place. We have not known the purpose of this research at this stage, so writing what the authors have chosen is clumsy at this point.
L52-53, shell and capsule is a totally different story so using these terms as synonyms is not correct.
L54 Could you please clarify what exactly means: "...their use allows for the localization of the site of the release..."?
L58 Before you write about auxiliary substances, please write about the basic substances that built the capsule.
It may seem obvious, but I think it would be good if the authors explain what the lyophilization process is about before giving the examples of application in the industry.
Please add information whether and which lyophilized fruit and / or vegetable preparations are currently available on the world and / or European market.
Materials and methods:
Please specify the period (month, year) the apples were harvested and the time and conditions they were stored after harvesting.
L328 “similarly to the technique described in previous studies” if the research was carried out in the same conditions as in the reference [45,46], it's okay to give the source, but if the authors write "similarly", they must describe exactly what modifications they made to the methodology.
But what were these capsules? where did they come from? what was their composition? how were they obtained?
“One capsule was weighed, its content was poured out, and then the capsule shell was weighed.” - what content? please, add more specific information about capsules.
Results and discussion:
L85-149, Fine, so the authors got the results they expected, they confirmed what was already researched, described and published. Please indicate the novelty in this research procedure, or prove that it was necessary in the planned experiment. It is not entirely clear to me whether the authors discuss their results with the results of other authors who have tested apples or lyophilized preparations.
I believe that the variation in the composition of the capsules requires a more comprehensive description.
General remark:
The standard deviation presented in the column charts is not clear, it seems that in some cases these values are the same and minimal. I kindly ask the authors to comment on this issue.
In my opinion, the discussion is poorly described. It is necessary to thoroughly discuss the results compared to the manuscripts of other authors.
Please indicate the similarities and differences in these works in compare to other publications, please underline what the innovative approach was.
The general remark is that despite properly conducted research, it is nothing novel in the research approach. Yes, the research assumption and the idea of creating a freeze-dried apple enclosed in a capsule is interesting, but I do not see anything new in the methods used in this study.
I am wondering if the authors choose the right journal to publish their research.
Author Response
Dear Authors of manuscript entitled “Modeling and biopharmaceutical evaluation of capsules with lyophilized apple powder”, below please find my comments and suggestions: 1. Abstract - please indicate strongly the aim of your research. In addition, please add information on how the lyophylisate was obtained and what the capsule was made of (it has a big impact on the results, so it is worth mentioning it at the beginning). Writing "capsules of different compositions" is too generate even for Abstract. Maybe it would be helpful for potential readers to add kay words such as "lyophylysat" and/or "capsule"? Thank you for the remark. In the abstract and introduction section, we changed the aim to "The aim of our study was to model hard gelatin capsules with lyophilized apple powder by using different excipients and to evaluate the release kinetics of phenolic compounds ". We supplemented information on the lyophilization process and what the capsules were made of (L 12-15): "The apple slices of 'Ligol' cultivar were immediately frozen in a freezer (at -35°C) with air circulation and were lyophilized with a sublimator at the pressure of 0.01 mbar (condenser temperature, -85°C). Lyophilized apple powder was used as an active material filled into hard gelatin capsules". In terms of "capsules of different compositions" we had in mind (L15 - 17) "apple lyophilizated-containing capsules of different encapsulating content".
Introduction - More recent reports (than from 2014, L31) on the cultivation and consumption of apples are necessary.
Thank you for your attentiveness, we changed report in to 2020 years information.
L 38-39, please provide the average amount/concentration of mentioned compounds.
Thank you for the remark. We adduce the data about amount of mentioned biological active compounds (L40-48): "They have been found to contain a complex of phenolic (8.2–360.75 mg/g-1) [5], and triterpenic (0.47 ̶ 3.75 mg/g-1) [6] compounds, glucose (1.96 ̶ 75.91 mg/g-1), fructose (15.52 ̶ 342.14 mg/g-1), sucrose (11.81 ̶ 43.79 mg/g-1), xylitol (1.01 ̶ 13.79 mg/g-1), pectins (4.05 ̶ 19.72 mg/g-1), organic acids (malic acid (38.96 ̶ 107.28 mg/g-1), citric acid (1.84–4.12 mg/g-1), maleic acid (0.17 ̶ 0.23 mg/g-1), pyruvic acid (0.14–0.34 mg/g-1), shikimic acid (0.04 ̶ 0.17 mg/g-1)) [7,8], vitamins (21.08 mg/g-1) [9], macroelements (K (1.07–1.12 mg/g-1), P (0.074–0.11 mg/g-1), Mg (0.05–0.08 mg/g-1), Ca (0.04–0.06 mg/g-1), Na (0.007–0,04 mg/g-1)), microelements (Fe (0.001–0.003 mg/g-1), Zn (0.0004–0.019 mg/g-1), Mn (0.0003–0.0004 mg/g-1), Cu (0.0003–0.0005 mg/g-1) [10], and fiber (3700–4500 mg/g-1) [5] ".
Apples are great and we should eat them, but maybe it is worth mentioning that eating these fruits in excess is not advisable and is associated with unpleasant ailments? (fermentation, flatulence). Maybe this suggests that it would also be good to consume a lyophilisate, e.g. for someone who has flatulence after eating apples? Please treat this comment as a soft, non-binding remark. L51-52, in my opinion the sentence: “In our study, we chose capsules as a solid pharmaceutical form” is useless in this place. We have not known the purpose of this research at this stage, so writing what the authors have chosen is clumsy at this point.
Thank you for your attentiveness. The sentence in lines 51-52 has been deleted.
L52-53, shell and capsule is a totally different story so using these terms as synonyms is not correct.
Thank you for your comment. In the text we uniformed and used terms "capsule".
L54 Could you please clarify what exactly means: "...their use allows for the localization of the site of the release..."?
One of the most widely used capsules may not be resistant to and dissolved in gastric juice, so we decided used this capsules type in the study. Also, the capsules may be resistant to gastric juice and dissolved in a selected part of the gastrointestinal tract if appropriate excipients are selected. Therefore, one of the most important advantages of capsules is that their use allows for the localization of the gastrointestinal tract site of the release of biologically active compounds from the capsule, thus helping to protect the acid-sensitive biologically active substances from the destructive effects of gastric juice.
L58 Before you write about auxiliary substances, please write about the basic substances that built the capsule. It may seem obvious, but I think it would be good if the authors explain what the lyophilization process is about before giving the examples of application in the industry. Please add information whether and which lyophilized fruit and / or vegetable preparations are currently available on the world and / or European market. We agree with your comment.
In our study, the basic substance in capsules is a lyophilized apple powder rich in phenolic compounds. We explain the lyophilization process (L 83-97): "About 50% of the currently biopharmaceuticals are lyophilized, representing the most common formulation strategy [17]. Lyophilization or freeze drying is a process in which water is frozen, followed by its removal from the sample, initially by sublimation (primary drying) and then by desorption (secondary drying). Freeze-drying is a process of drying in which water is sublimed from the product after it is frozen [17, 18] " and added information of lyophilized fruit and / or vegetable (L92-97): "Lyophilized fruits (apples, pears, plums, peaches), berries (blueberries, cranberries, strawberries), and vegetables (carrots, beets) used in the food industry and can be added to various types of cocktails, cereals, drinks, meals, juices, yogurts, ice cream, freeze-dried fruit powder can also be added to food supplements thus increasing the content of biological active compounds in prepared [19-22]. Dehydrated, freeze-dried fruit, berries, vegetables promote long-term storage of freeze-dried food or food supplements".
Materials and methods: Please specify the period (month, year) the apples were harvested and the time and conditions they were stored after harvesting.
The apples harvested in September 2019, were immediately lyophilized and used for the study.
L328 “similarly to the technique described in previous studies” if the research was carried out in the same conditions as in the reference [45,46], it's okay to give the source, but if the authors write "similarly", they must describe exactly what modifications they made to the methodology.
Thank you for the remark. We described in more detail Preparation of phenolic extracts (L400-406) and Qualitative and quantitative analysis by HPLC-PDA method (L 409-418) sections: Preparation of phenolic extracts. During the analysis of phenolic compounds, 2.5 g of lyophilizate powder (exact weight) was weighed, added to 30 mL of 70% (v/v) ethanol, and extracted in a Sonorex Digital 10 P ultrasonic bath (Bandelin Electronic GmbH & Co. KG, Berlin, Germany) at room temperature for 20 min. The obtained extract was filtered through a paper filter, and the residue on the filter was washed with 70% (v/v) ethanol in a 50 mL flask until the extract volume was reached. The conditions of the extraction were chosen based on the results of the tests for setting the extraction conditions [65]. Qualitative and Quantitative Analysis by HPLC-PDA Method. The qualitative and quantitative HPLC analysis of phenolic compounds was performed with a Waters 2998 PDA detector (Waters, Milford, USA). Chromatographic sepa-rations were carried out by using a YMC-Pack ODS-A (5 μm, C18, 250×4.6 mm i.d.) column. The column was operated at a constant temperature of 25 °C. The volume of the analyzed extract was 10 μL. The flow rate was 1 mL/min. The mobile phase consisted of 2% (v/v) acetic acid (solvent A) and acetonitrile (solvent B). Gradient variation: 0-30 min 3-15% B, 30-45 min 15-25% B, 45-50 min 25-50% B, and 50-55 min 50-95% B. For the quantitative analysis, the calibration curves were obtained by injecting the known concentrations of different standard compounds. All the identified phenolic compounds were quantified at ?=200-400 nm wavelength [66, 67].
But what were these capsules? where did they come from? what was their composition? how were they obtained? “One capsule was weighed, its content was poured out, and then the capsule shell was weighed.” - what content? please, add more specific information about capsules.
Compositions of capsules fillings are given in Table 2. Powder was prepared by simple mixing of the mixture of apple lyophilizate with excipient. Filled capsules were pre-pared using the manual capsule filling machine (Capsuline, Florida, USA).We corrected the sentence: "One capsule was weighed, apple lyophilizate and excipients mixture were poured out, and then the capsule shell was weighed".
Results and discussion: L85-149, Fine, so the authors got the results they expected, they confirmed what was already researched, described and published. Please indicate the novelty in this research procedure, or prove that it was necessary in the planned experiment. It is not entirely clear to me whether the authors discuss their results with the results of other authors who have tested apples or lyophilized preparations. I believe that the variation in the composition of the capsules requires a more comprehensive description. Thank you for the comment. The text was supplemented the other authors' research results and compared with our study results.
L 119 Added sentence: "The data obtained by Jakobec et al. corroborate the results of our research [23] ".
L129-131 Supplemented by data: "Studies of the fruit samples of apple cultivars grown in the orchards of the Marche region of Italy showed that the total amount of phenolic compounds ranged from 0.82 mg/g to 3.60 mg/g and confirms the results of our research [5] ".
L 140-143 Added information: "The amount of chlorogenic acid in apple sample grown in Italy was found to vary from 0.12 mg/g to 0.63 mg/g [26]. In our study was determined the higher amount of chlorogenic acid compared with amount described of Italian scientists".
L 153-155 Added data: "The flavan-3-ol content in fruit samples of apple cultivars grown in Croatian orchards was found to vary from 0.02 mg/g to 0.69 mg/g [23]. We established the higher amount of total flavan-3-ols compared with amount determined by Croatian scientists".
L160-162 Added data: " The amount of procyanidin B2 and procyanidin C1 in fruit samples of apple cultivars grown in Polish orchards ranged from 0.07 mg/g to 2.00 mg/g and 0.0006 mg/g to 0.97 mg/g, accordingly [37]. Our study results confirmed Polish research results".
L 172-174 Supplemented by information: " The amount of phloridzin in fruit samples of apple cultivars grown in orchards in the Garfagnana region of Italy ranged from 0.01 mg/g to 0.05 mg/g [44]. The results obtained in these studies confirm the results obtained in our research".
L 184-186 Added data: " Studies of fruit samples of apple cultivars grown in Croatian orchards showed that quercetin glycosides levels ranged from 0.20 mg/g to 1.22 mg/g [23]. The results of our study are corroborated by research data obtained by Croatian researchers".
L 189-191 Added data: " The amount of hyperoside in fruit samples of apple cultivars grown in Italian orchards ranged from 0.0003 mg/g to 0.002 mg/g [47]. The apple lyophilisate contained higher amounts of hyperoside compared to these found in fruit samples of apple cultivars grown in Italian orchards. ".
L 192-195 Supplemented by information: " The amount of quercitrin was 0.09 ± 0.04 mg/g and these results of study confirmed Italian researches study results described that the amount of quercitrin found in apple samples ranged from 0.005 mg/g to 0.043 mg/g [47] ".
L 247-251 Supplemented by information: " The results of our study were confirmed other scientists described results. Vyas et al. study results showed, that lyophilized Vasa Swaras capsules disintegration more than 4.23 minutes [53]. Esmaeili et al. confirmed our study results, that hard gelatin capsules from Pinus eldarica bark extract disintegration between 9.4 to 20.0 minutes [54]". L 318-327 Added by information: "Literature has demonstrated concerning data related to herbal medicinal product dissolution tests. A study performed with phytomedicines based on Ginkgo biloba extract capsules and tablets, marked differences in dissolution behavior were established, with values of 99% and 33% dissolution, on average, after 15 minutes and 60 minutes, respectively [57]. The dissolution profile of Senna sp. was less than 10% sennoside release from capsules containing dry extract, in a period of 60 minutes, in contrast to lyophilized Senna sp. extract, which attained around 90% dissolution after the same period. Data on the dissolution test of Passiflora sp. showed that capsules containing the crude extract presented 50% dissolution while lyophilized extract and other stand-ardized extract reached around 100% for the same period [58]". L 351-355 Supplemented by information: "According to the scientific literature, lyophilisate of Vasa Swaras capsules improved the stability and oral bioavailability of vasicine as well as reduced its conversion to vasicinone. The pharmacokinetic profiles of the various formulations inferred that the highest bioavailability of vasicine was obtained from hard gelatin capsules of lyophilized Vasa Swaras [53].".
General remark: The standard deviation presented in the column charts is not clear, it seems that in some cases these values are the same and minimal. I kindly ask the authors to comment on this issue. In my opinion, the discussion is poorly described. It is necessary to thoroughly discuss the results compared to the manuscripts of other authors. Please indicate the similarities and differences in these works in compare to other publications, please underline what the innovative approach was. The general remark is that despite properly conducted research, it is nothing novel in the research approach. Yes, the research assumption and the idea of creating a freeze-dried apple enclosed in a capsule is interesting, but I do not see anything new in the methods used in this study. I am wondering if the authors choose the right journal to publish their research Thank you for the remark. The standard deviation is very small or no different in some cases, as the differences between the results obtained after three replicates are small.
We supplemented the text with data from other researchers and compared the results of their research with the results of our study. Added sentences: L 119, L129-131, L 140-143, L 153-155, L160-162, L 172-174, L 184-186, L 189-191, L192-195, L 247-251, L 318-327, L 351-355. Considering the increasing consumption of herbal medicines, food supplements, there is a need for studies to ensure quality from raw material to finished product. Quality control is essential for the safety of function food or nutrition supplements. Thus, biopharmaceutical evaluation of capsules with apple lyophilisate studies are of great importance since they can contribute toward ensuring futures researches, planning innovative nutrition supplements manufacture.

Reviewer 2 Report
The paper is presenting an interesting study of biopharmaceutical evaluation of the apple lyophilisate powder with the focus on the polyphenols’ release. The research is well planned, the results are interesting. However, there are some details that authors should improve for the results to be better presented. The methods section would have to be more precise, data about extraction procedure and HPLC analysis are missing. The second major point is about expression of the results. They should on my opinion be expressed in mg/g of lyophilized apple, not per mL of extract, which is not really relevant and the explanation of extraction procedure is not sufficient anyhow. Like that, the values can be compared much more easily. And finally the discussion is rather poor, it is lacking the comparisment to other similar more recent studies.
Some other more specific comments:
Line 83: apple sample: change to “apple lyophilisate (change also in the methods)
Line 96: explain which peaks are determined at which wavelength
Line 101-103: It is also not too clear how the total mass of polyphenols was determined.
Table 3: please specify what are the numbers representing ad what are the units
Figure 5: There is no need to present data in 4 separated graphs and even less putting the legends for each polyphenolic compounds 4 times.
Figure 6a should be presented individually, Figure 6 b, c, d should be removed
Lines 242, 243: not clear sentence
Line 290: please revise the sentence
Line 322: change to “Preparation of phenolic extracts” and briefly explain the procedure
Line 328: specify the detector and specify the separation conditions as well as the number of repetitions performed for each sample
Author Response
The paper is presenting an interesting study of biopharmaceutical evaluation of the apple lyophilisate powder with the focus on the polyphenols’ release. The research is well planned, the results are interesting. However, there are some
details that authors should improve for the results to be better presented. The methods section would have to be more precise, data about extraction procedure and HPLC analysis are missing. The second major point is about expression of the results. They should on my opinion be expressed in mg/g of lyophilized apple, not per mL of extract, which is not really relevant and the explanation of extraction procedure is not sufficient anyhow. Like that, the values can be compared much more easily. And finally the discussion is rather poor, it is lacking the comparisment to other similar more recent studies.
Thank you for the remark. In figure 2 and figure 3 data units changed to mg/g.
Some other more specific comments:
Line 83: apple sample: change to “apple lyophilisate (change also in the methods)
Thank you for the remark. The word “apple sample” has been replaced with to “apple lyophilisate”.
Line 96: explain which peaks are determined at which wavelength
In chromatogram of the ethanol extract of the apple lyophilisate analytes determined at λ=280 nm wavelength: 1 – procyanidin B1; 2 – (+)-catechin; 3 – chlorogenic acid; 4 – procyanidin B2; 5 – ( ̶ )-epicatechin; 6 – procyanidin C1; At
λ=360 nm wavelength: 7 – rutin; 8 – hyperoside; 9 – isoquercitrin; 10 – reynoutrin; 11 – avicularin; 12 – quercitrin; At λ=280 nm wavelength 13 – phloridzin.
Line 101-103: It is also not too clear how the total mass of polyphenols was determined.
In this paper, the total amount of phenolic compounds in apple lyophilisate was the sum of the identified and quantified individual phenolic compounds. We limited the analysis to the identified phenolic compounds and presented their arithmetic sum.
Table 3: please specify what are the numbers representing ad what are the units
Thank you for your attentiveness. In table 3, we inserted row with concentration units (μg/ml).
Figure 5: There is no need to present data in 4 separated graphs and even less putting the legends for each polyphenolic compounds 4 times.
Thank you for the remark. Figure 5 has been replaced with Table 4.
Figure 6a should be presented individually, Figure 6 b, c, d should be removed.
Thank you for the remark. Figure 6 has been replaced with Table 5.
Lines 242, 243: not clear sentence.
The sentences in lines 242-243 was corrected.
Line 290: please revise the sentence
The sentences in line 290 was correcet and abbreviated to: “The study included 'Ligol' apple cultivar (a winter cultivar bred in Poland) ”.
Line 322: change to “Preparation of phenolic extracts” and briefly explain the procedure
Thank you for the remark. We changed into “Preparation of phenolic extracts” and suplemented information of extraction procedure: “During the analysis of phenolic compounds, 2.5 g of lyophilizate powder (exact weight) was weighed, added to 30 mL of 70% (v/v) ethanol, and extracted in a Sonorex Digital 10 P ultrasonic bath (Bandelin Electronic GmbH & Co. KG, Berlin, Germany) at room temperature for 20 min. The obtained extract was filtered through a paper filter, and the residue on the filter was washed with 70% (v/v) ethanol in a 50-mL flask until the extract volume was reached. The conditions of the extraction were chosen based on the results of the tests for setting the extraction conditions [65] ”.
Line 328: specify the detector and specify the separation conditions as well as the number of repetitions performed for each sample
Thank you for the remark. We described in more detail about Qualitative and quantitative analysis by HPLC-PDA method : “The qualitative and quantitative HPLC analysis of phenolic compounds was performed with a Waters 2998
PDA detector (Waters, Milford, USA). Chromatographic separations were carried out by using a YMC-Pack ODS-A (5 μm, C18, 250×4.6 mm i.d.) column. The column was operated at a constant temperature of 25 °C. The volume of the analyzed extract was 10 μL. The flow rate was 1 mL/min. The mobile phase consisted of 2% (v/v) acetic acid (solvent A) and acetonitrile (solvent B). Gradient variation: 0-30 min 3-15% B, 30-45 min 15-25% B, 45-50 min 25-50% B, and 50-55 min 50-95% B. For the quantitative analysis, the calibration curves were obtained by injecting the known concentrations of different standard compounds. All the identified phenolic compounds were quantified at ?=200-400 nm wavelength [66,67] ”.

Reviewer 3 Report
This article is a biopharmaceutical study on phenolic compounds evaluation of hard gelatin capsules with lyophilized apple powder. Moreover, they performed tests on capsule disintegration and dissolution to evaluate the quality of apple lyophilizates contained in capsules of different compositions.
The topic of paper is interesting. However, paper requires major revision because needs some additional work and information to be considered for publication.
Specific comments and suggestions for improving the paper are:
Title, I suggest to delete Modeling
Line 10-12, please, delete the sentence “They have be found….and fiber”.
Line 79, please, specified better this sentence: “that would provide the body” which body?… human body?
Figure 1, congratulations for the chromatogram with the chemical structure of the phenolic compounds! Please, in the caption specify the wavelength of that chromatogram…is it 280 or 360 nm?
Table 3: please, check the letters and insert A for the higher number and in decreasing number B and C…
Table 3, please, insert a row with μg/ml
Line 101, please, specify 183.07…..Figure 2 and Figure 3?
Line 199, please, delete that
Figure 5: I suggest to delete this figure and insert a table
Figure 6, please, as above
Line 364-365, please, rewrote better this sentence…
Line 372-373, please, as above
Author Response
This article is a biopharmaceutical study on phenolic compounds evaluation of hard gelatin capsules with lyophilized apple powder. Moreover, they performed tests on capsule disintegration and dissolution to evaluate the quality of apple lyophilizates contained in capsules of different compositions. The topic of paper is interesting. However, paper requires major revision because needs some additional work and information to be considered for publication. Specific comments and suggestions for improving the paper are:
Title, I suggest to delete Modeling
Thank you for the remark. We deleted “Modeling”.
Line 10-12, please, delete the sentence “They have be found….and fiber”.
Thank you for your attentiveness. The sentence in lines 10-12 has been deleted.
Line 79, please, specified better this sentence: “that would provide the body” which body?… human body?
The sentences in line 79 was corrected and replaced into: “The aim of our study was to model hard gelatin capsules with lyophilized apple powder by using different excipients and to evaluate the release kinetics of phenolic compounds”. Figure 1, congratulations for the chromatogram with the chemical structure of the phenolic compounds! Please, in the caption specify the wavelength of that chromatogram…is it 280 or 360 nm?
In chromatogram of the ethanol extract of the apple lyophilisate analytes determined at λ=280 nm wavelength: 1 – procyanidin B1; 2 – (+)-catechin; 3 – chlorogenic acid; 4 – procyanidin B2; 5 – ( ̶ )-epicatechin; 6 – procyanidin C1; At λ=360 nm wavelength: 7 – rutin; 8 – hyperoside; 9 – isoquercitrin; 10 – reynoutrin; 11 – avicularin; 12 – quercitrin; At λ=280 nm wavelength 13 – phloridzin.
Table 3: please, check the letters and insert A for the higher number and in decreasing number B and C… Table 3, please, insert a row with μg/ml
Thank you for your attentiveness. In table 3, we inserted row with concentration units (μg/ml) and inserted the higher letters with decreasing letters in column.
Line 101, please, specify 183.07…..Figure 2 and Figure 3?
In this paper, the total amount of phenolic compounds in apple lyophilisate was the sum of the identified and quantified individual phenolic compounds. We limited the analysis to the identified phenolic compounds and presented their arithmetic sum. In figure 2 and figure 3 showed the amount of individual phenolic compounds.
Line 199, please, delete that
Thank you for your attentiveness. The sentence in line 199 has been deleted. Figure 5: I suggest to delete this figure and insert a table
Thank you for the remark. Figure 5 has been replaced with Table 4.
Figure 6, please, as above
Thank you for the remark. Figure 6 has been replaced with Table 5.
Line 364-365, please, rewrote better this sentence…
Thank you for your attentiveness. The sentence in lines 364-365 has been deleted.
Line 372-373, please, as above
Thank you for your attentiveness. The sentence in lines 272-273 has been deleted.

Round 2
Reviewer 1 Report
Dear Authors,
Thank you for improving the quality of your research adding more information and following reviewer's comments. I am still think that proposed manuscript is lack of novelty and not exactly fits to this journal, however in present form it is good organised and well prepared.
Author Response
Thank you for the remark. We are really glad to correct the manuscript of the article based on your comments.
Reviewer 2 Report
The authors have resolved most of the issues, raised in the initial review. Please just change still Figure 2 and 3 with results, expressed in corrent units (mg/g).
Author Response
Thank you for the remark. We have provided units of measurement expressed in mg / g in Figure 2 and 3.
Reviewer 3 Report
Please check table 3!
Please in the row once the unit of measurement and center it! Please delete concentration!
Not decreasing letters in column…..decreasing letter in relation to numeric values! Please, insert A for the higher number… for example in the first column:.
|
372.5b |
|
367.5b |
|
372.5b |
|
367.5b |
|
385.0b |
|
589.5a |
|
640.3a |
|
167.9c |
|
126.4c |
|
68.4d |
Please correct the other columns!
Author Response
Thank you for the remark. In the row of Table 3, the units were centered and the concentration was deleted. We've corrected the letters based on your comment. The letter A was inserted next to the highest number and the other letters B, C, D accordingly.